
# General quantum-classical dynamics
# as measurement based feedback

Antoine Tilloy

Laboratoire de Physique de l'Ecole Normale Supérieure, Mines Paris - PSL,
CNRS, Inria, PSL Research University, Paris, France

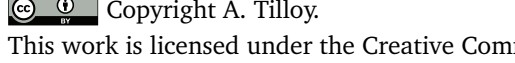

antoine.tilloy@minesparis.psl.eu

## Abstract

This note derives the stochastic differential equations and partial differential equation of general hybrid quantum–classical dynamics from the theory of continuous measurement and general (non-Markovian) feedback. The advantage of this approach is an explicit parameterization, without additional positivity constraints. The construction also neatly separates the different effects: how the quantum influences the classical and how the classical influences the quantum. This modular presentation gives a better intuition of what to expect from hybrid dynamics, especially when used to construct possibly fundamental theories.

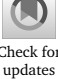

# 1   Introduction

It is possible to construct consistent hybrid models where quantum and classical variables evolve jointly. This can be used, for example, to construct theories in which a classical gravitational field is coupled with quantum matter. The price to pay for this hybrid coupling is only extra randomness in the dynamics. The conceptual benefit, apart from allowing quantum–classical dynamics in the first place, is the emergence of a spontaneous collapse process that can help solve the measurement problem.

Formally, such models are less exotic than they would seem: requiring that the dynamics is Markovian and continuous yields the same dynamical equations as continuous quantum measurement combined with feedback. In fact, this is precisely how hybrid models were constructed in the context of Newtonian gravity [1–4] and in even earlier works [5], before the generality of the approach was appreciated [6]. The objective of this technical note is to explicitly construct the general equations of quantum–classical dynamics from continuous measurement and feedback. This provides a powerful reinterpretation of recent works. The only non-trivial starting point is the stochastic master equation (SME) of continuous quantum measurement, which has been developed in quantum optics and quantum foundations since the late eighties [7, 8], and is now in standard textbooks [9, 10]. From this direct stochastic description, a partial differential equation (PDE) à la Fokker-Planck (as used *e.g.* by Oppenheim [11] in more recent developments) can be re-derived. Provided one is familiar with Itô calculus, or willing to use the simple rules given in the appendix of the present paper, this gives the shortest and most intuitive path to continuous hybrid dynamics.

In my opinion, presenting the formalism this way comes with a number of additional benefits, regardless of one's familiarity with quantum optic techniques:

1. One can leverage well understood equations and carry the derivations rigorously directly in the continuum.

2. The models one obtains are parameterized in a natural and constructive way. More precisely, there are no extra constraints that need to be fulfilled for the models to be consistent, and yet the parameterization generates *all* that is possible.

3. The measurement and feedback formulation provides a very powerful intuition pump about the physics, allowing to essentially guess most results without computations. Its merit is its modularity, neatly separating the various contributions to the dynamics: the intrinsic quantum and classical dynamics, the classical acting on the quantum, and the quantum acting on the classical.

4. The link with spontaneous collapse models is transparent, allowing to recycle lessons acquired in this field, in particular regarding spontaneous heating.

In this presentation of hybrid dynamics from measurements, the *measurement signal* plays a central role, as the glue between the quantum and classical. This is a useful quantity (conceptually and practically) that does not appear naturally if one starts from general hybrid dynamics written as a partial differential equation.

Before diving into the equations, it is important to warn the reader. We will see how quantum classical dynamical models can generically be obtained from the theory of continuous measurement and feedback. This does not mean that one should take this interpretation literally, especially if one is in the business of building models of Nature that are possibly fundamental! We do not expect that something in Nature *actually* carries measurement and feedback. The equivalence is, of course, only mathematical. This is sufficient to give us access to a well developed toolbox of techniques and a powerful intuition pump: we usually have a

better idea of what to expect from measurement and feedback than from general quantum–classical dynamics.

## 2 Continuous measurement and feedback

### 2.1 The stochastic master equation for continuous measurement

Our starting point is the standard stochastic master equation (SME) in Itô form (see appendix for a quick explanation of Itô calculus), describing the *continuous* evolution of a quantum system of density matrix $\rho$ under continuous measurement. Continuous measurement can also yield quantum state dynamics with discrete jumps, but we restrict the analysis in this note to continuous (diffusive) dynamics (see remark 6). For a system with density matrix $\rho$, which one can take finite dimensional for simplicity, the most general continuous SME is

$$\mathrm{d}\rho = -i[H_0, \rho]\,\mathrm{d}t + \sum_{k=1}^{n} \mathcal{D}[\hat{c}_k](\rho)\,\mathrm{d}t + \sqrt{\eta_k}\,\mathcal{M}[\hat{c}_k](\rho)\,\mathrm{d}W_k\,, \tag{1}$$

where the $W_k$ are $n$ independent Wiener processes (or Brownian motions), $0 \leq \eta_k \leq 1$ are the *detector efficiencies*, and

$$\mathcal{D}[\hat{c}](\rho) = \hat{c}\rho\hat{c}^\dagger - \frac{1}{2}\left\{\hat{c}^\dagger\hat{c}, \rho\right\} \quad \texttt{[dissipation / decoherence]}, \tag{2}$$

$$\mathcal{M}[\hat{c}](\rho) = \hat{c}\rho + \rho\hat{c}^\dagger - \mathrm{tr}\left[(\hat{c}+\hat{c}^\dagger)\rho\right]\rho \quad \texttt{[stochastic innovation]}. \tag{3}$$

The operators $\hat{c}_k$ are *arbitrary* (not necessarily commuting, not necessarily Hermitian) characterizing each detector, and $H_0$ is a Hermitian operator specifying some pre-existing unitary dynamics. The corresponding "sharp" measurement signals $I_k(t) = \frac{\mathrm{d}r_k(t)}{\mathrm{d}t}$ verify:

$$\mathrm{d}r_k = \frac{1}{2}\mathrm{tr}[(\hat{c}_k + \hat{c}_k^\dagger)\rho]\,\mathrm{d}t + \frac{1}{2\sqrt{\eta_k}}\mathrm{d}W_k\,. \tag{4}$$

This signal equation (4) can be used to express $\mathrm{d}W_k$ as a function of $\mathrm{d}r_k$ and thus to reconstruct $\rho$ as a function of the signal using (1)

$$\mathrm{d}\rho = -i[H_0, \rho]\,\mathrm{d}t + \sum_{k=1}^{n} \mathcal{D}[\hat{c}_k](\rho)\,\mathrm{d}t + \eta_k\,\mathcal{M}[\hat{c}_k](\rho)\left(2\,\mathrm{d}r_k - \mathrm{tr}\left[(\hat{c}_k + \hat{c}_k^\dagger)\rho\right]\mathrm{d}t\right). \tag{5}$$

Numerically, this latter equation (5) is typically used to reconstruct quantum trajectories from experimental measurements, whereas equation (1) is used if one wants to sample quantum trajectories directly, and numerically generate a signal with the correct law. The continuous SME (1) has been known since the eighties, and independently invented and reinvented in mathematics [8, 12], quantum optics [13], and foundations [7] (see *e.g.* Jacobs and Steck for pedagogical introduction [14]). The simplest way to obtain it is as the limit of infinitely weak and infinitely frequent measurements (see for example [15–17]). We may however forget its origin, and simply use it as a provably consistent starting point.

*Remark* 1 (Diagonal form). We have specified the measurement dynamics in its so called "diagonal form". In particular, the deterministic part of the dynamics is written:

$$\mathcal{L}(\rho) = -i[H_0, \rho] + \sum_k \hat{c}_k\rho\hat{c}_k^\dagger - \frac{1}{2}\left\{\hat{c}_k^\dagger\hat{c}_k, \rho\right\}, \tag{6}$$

which, historically, is the form introduced by Lindblad [18]. An equivalent representation is the non-diagonal form, which was introduced at the same time by Gorini, Kossakowski, and Sudarshan [19]

$$\mathcal{L}(\rho) = -i[H_0, \rho] + \sum_{\alpha, \beta} D_{\alpha\beta} \left( \hat{L}_\alpha \rho \hat{L}_\beta^\dagger - \frac{1}{2} \left\{ \hat{L}_\beta^\dagger \hat{L}_\alpha, \rho \right\} \right), \tag{7}$$

which seems more general but is equivalent (which is seen by diagonalizing $D$) provided the necessary semi-definite positivity constraint $D \succeq 0$ is enforced. This non-diagonal form extends to the complete measurement SME, and we write it explicitly in (19). There, the equation may also seem more general but is in fact equivalent because of the positivity constraint.

*Remark* 2 (Regularity). Technically, the sharp signals are only defined as distributions, and one should define directly the "smooth" signal $I_{k,\varphi}$ corresponding to the sharp signal integrated against a smooth function $\varphi$

$$I_{k,\varphi} := \int \varphi(t) \, dr_k(t) = " \int \varphi(t) I_k(t) \, dt \ ". \tag{8}$$

Interestingly, before we turn on a general feedback, the correlation functions of the signal

$$\mathbb{E}\big[ I_{k_1, \varphi_1} I_{k_2, \varphi_2} \cdots I_{k_N, \varphi_N} \big], \tag{9}$$

where $\mathbb{E}$ denotes the average over the noise / signal, can be computed exactly [20, 21] for finite dimensional Hilbert spaces (and well approximated otherwise).

*Remark* 3 (Efficiencies). If $\forall k$, $\eta_k = 1$, the SME preserves pure states $\rho = |\psi\rangle\langle\psi|$ and could thus be rewritten as a stochastic Schrödinger equation (SSE) for $|\psi\rangle$. Experimentally, the efficiencies can be quite low, but if we are constructing fundamental models it makes sense to fix them at 1.

*Remark* 4 (Non-linearity). The SME is non-linear but this non-linearity is mild, and is purely an effect of the fixed normalization $\mathrm{tr}(\rho) = 1$. To see this, one can introduce $\tilde{\rho}$ the un-normalized density matrix, which is equal to $\rho$ at the initial time, and which verifies the *linear* SME (as a function of the signal)

$$d\tilde{\rho} = -i[H_0, \tilde{\rho}] \, dt + \sum_{k=1}^{n} \mathcal{D}[\hat{c}_k](\tilde{\rho}) \, dt + 2\eta_k \left( \hat{c}_k \tilde{\rho} + \tilde{\rho} \hat{c}_k^\dagger \right) dr_k. \tag{10}$$

Using Itô's lemma, one verifies that $\rho_t = \frac{\tilde{\rho}_t}{\mathrm{tr}(\tilde{\rho}_t)}$.

*Remark* 5 (Collapse models). Mathematically, for $\hat{c}_k$'s taken to be regularized mass density operators, the SME (1) is *exactly* the stochastic differential equation of a collapse (or spontaneous localization) model [22, 23]. The SME (1) can even reproduce so called dissipative collapse models [24] by taking general non self-adjoint $\hat{c}_k$'s. Hence, whether we like it or not, hybrid quantum–classical models do contain a general collapse model at their core.

In an actual experiment, everything one can know about the system is stored in the signal trajectories $r_k(t)$. The rest, like the state $\rho_t$ itself, is not directly observable, and is only *reconstructed* from $r_k$. Practically, the signal is thus a crucial object, if only because it is the only thing we have. This is at the very least a hint that it is also a relevant quantity if we are in the business of building models.

## 2.2 General feedback

At this stage, we already have a proto-hybrid model, in the sense that we have a classical signal $r_k$ sourced from a quantum a part $\rho$. What remains is to have the classical part back-react on the quantum part. To this end, we may enrich the classical part with a $d$-dimensional vector of classical variables $\mathbf{z} = \{z_a\}_{a=1\cdots d}$ depending on the measurement signals. Intuitively, in the experimental context, this vector of variables $\mathbf{z}$ can be whatever function of past results which is used in the control loop (the internal state of the controller). Again, if we want to build a fundamental model, this interpretation is not literal, but merely an intuition pump to guess the results.

Because it has the regularity of white noise, the signal can only enter linearly in the dynamics, and thus the most general adapted Itô process one can construct for $\mathbf{z}_t$ is

$$\mathrm{d}z_a = F_a(\mathbf{z})\,\mathrm{d}t + \sum_{k=1}^{n} G_{ak}(\mathbf{z})\,\sqrt{\eta_k}\,\mathrm{d}r_k \tag{11}$$

$$= F_a(\mathbf{z})\,\mathrm{d}t + \sum_{k=1}^{n} \frac{G_{ak}(\mathbf{z})\sqrt{\eta_k}}{2}\mathrm{tr}[(\hat{c}_k + \hat{c}_k^\dagger)\rho]\,\mathrm{d}t + \frac{G_{ak}(\mathbf{z})}{2}\mathrm{d}W_k\,. \tag{12}$$

The deterministic $F_a(\mathbf{z})$ part includes Hamiltonian dynamics as a special case, but can in principle be more general (*e.g.* with classical dissipation).

In (12) we extracted $\sqrt{\eta_k}$ away to define $G$: this is only a notational convenience to make the feedback finite in the case where $\eta_k = 0$, to make the connection with the notations of Weller-Davis and Oppenheim [6] more transparent. For simplicity, we take $\mathbf{z}$ to be a vector of real variables, and thus $F$ and $G$ are also real.

Now that we have classical variables (controller variables) $\mathbf{z}$, we can have everything depend on them in real time. This "everything" includes the measurement setting, *i.e.* the operators $\hat{c}_k \to \hat{c}_k(\mathbf{z})$ themselves, the pre-existing unitary dynamics $H_0 \to H(\mathbf{z})$, and potentially even the efficiencies $\eta_k \to \eta_k(\mathbf{z})$. For simplicity, we will not always write their $\mathbf{z}$ dependence explicitly.

As we will see, the SME (1) combined with the classical dynamics (12) is ultimately equivalent to the hybrid equations of [6], and consequently provide the most general Markovian continuous quantum–classical evolution that one can write. Importantly, owing to the "diagonal" form we have worked with from the start, the dynamics trivially preserve complete positivity of $\rho$, no matter the value of the parameters. Consistency is built in from the start.

*Remark* 6 (Jumps). We have focused on dynamics continuous in time *and* Hilbert space, that is, without jumps. Time-continuous measurements with jumps also exist (which, experimentally, correspond to photo-detection instead of the homodyne readouts we have used so far). All the steps we have followed could in principle just as well be followed starting from such jump measurement dynamics. The resulting quantum–classical trajectories have been discussed very recently and in a mathematically rigorous manner by Barchielli [25]. In fact, the jump case corresponds precisely to the type of dynamics that were initially considered by Oppenheim [11]. The existence of fully continuous yet consistent quantum–classical dynamics is less expected, and easier to miss, hence why we focus on it in the present note. However, extending the analysis to the jump case, and better, to a mix of diffusive continuous evolution and jumps, would certainly be worthwhile.

# 3 Connection with continuous hybrid dynamics

## 3.1 The approach of Oppenheim and collaborators

Another orthogonal approach to derive the dynamics we have outlined is to consider the most general quantum dynamics preserving a classical sector. To this end, one first introduces a global density matrix $\varrho$ acting on $\mathcal{H}_{\text{tot}} := \mathcal{H}_{\text{quantum}} \otimes \mathcal{H}_{\text{classical}}$ such that

$$\varrho_t = \int d^d \mathbf{z}\, \rho_t(\mathbf{z}) \otimes |\mathbf{z}\rangle\langle\mathbf{z}|, \tag{13}$$

in a *fixed* basis $|\mathbf{z}\rangle$. The lack of coherences in $\mathcal{H}_{\text{classical}}$ implies that the variable $\mathbf{z} \in \mathbb{R}^d$ is classical.[1] One then tries to find the most general subset of Lindblad dynamics on $\varrho$ such that this diagonal structure on $\mathcal{H}_{\text{classical}}$ is preserved. This is, in a nutshell, the strategy followed by Oppenheim in his seminal paper [11],[2] as well as the one of earlier constructions of hybrid dynamics by Blanchard and Jaczik [26, 27].

The most general dynamics obtained in that manner yield two distinct classes of classical behavior, continuous or with jumps [28]. In the former case, which is the one we focus on here, one ultimately gets a fairly simple and explicit partial differential equation (PDE) for $\rho_t(\mathbf{z})$. This PDE can then be *unraveled* into a direct stochastic representation for classical variables jointly evolving with a quantum state according to coupled SDEs. The PDE can be seen as the Fokker-Planck description of the Langevin dynamics given by the SDEs.

The PDE and unraveled SDEs obtained this way are consistent, as long as the parameters specifying the dynamics verify a set of positivity conditions [28]. However, by a simple change of variable, (going from "non-diagonal" to "diagonal" form), one can see that the subset of consistent SDEs is exactly the same as the one we have derived from measurement and feedback.

To follow our objective of deriving everything from measurement and feedback, we now show how to recover the quantum–classical PDE from the measurement and feedback dynamics, via a straightforward application of Itô calculus. Then, we make the connection with the original parameterization of Oppenheim and collaborators explicit by breaking the diagonal form of the parameterization.

## 3.2 From stochastic to partial differential equation representation

In the Fokker-Planck (or PDE) representation, instead of having random variables $\rho_t$ and $\mathbf{z}_t$ evolving jointly, we have a deterministic equation for an un-normalized $\rho_t(\mathbf{z})$ heuristically given by (13). More rigorously, $\rho_t(\mathbf{z})$ is such that $\mathrm{tr}[\rho_t(\mathbf{z})]$ is the probability distribution $p_t(\mathbf{z})$ of $\mathbf{z}$ at time $t$, and $\rho_t = \rho_t(\mathbf{z}_t)/\mathrm{tr}[\rho_t(\mathbf{z}_t)]$. Mathematically, this means that $\rho_t(\mathbf{z})$ verifies

$$\forall f \in \mathcal{C}_0^\infty(\mathbb{R}^d), \quad \mathbb{E}[\rho_t f(\mathbf{z}_t)] = \int f(\mathbf{z}) \frac{\rho_t(\mathbf{z})}{\mathrm{tr}[\rho_t(\mathbf{z})]} p(\mathbf{z}) d^d\mathbf{z} = \int f(\mathbf{z}) \rho_t(\mathbf{z}) d^d\mathbf{z}. \tag{14}$$

---

[1]More precisely, given such a diagonal form, we may assume that $\mathbf{z}$ is measured at all time without disturbing the dynamics. Hence, there exists a natural classical process, which is just this measurement trajectory.

[2]Although this article [11] might seem like one of the most recent of this hybrid program, it is in fact an updated version of the first paper by Oppenheim, which appeared on arxiv 5 years earlier, in November 2018.

To find the partial differential equation obeyed by $\rho_t(\mathbf{z})$, a simple option,[3] is to differentiate the left-hand side:

$$
\begin{aligned}
\mathrm{d}\big[\rho_t f(\mathbf{z}_t)\big] = &-i f(\mathbf{z}_t)[H(\mathbf{z}),\rho]\,\mathrm{d}t + f(\mathbf{z}_t)\sum_{k=1}^{n}\mathcal{D}[\hat{c}_k](\rho)\,\mathrm{d}t + f(\mathbf{z}_t)\sqrt{\eta_k}\,\mathcal{M}[\hat{c}_k](\rho)\,\mathrm{d}W_k \\
&+ \frac{\partial f(\mathbf{z}_t)}{\partial z_a}F_a(\mathbf{z}_t)\rho_t\,\mathrm{d}t + \frac{\partial f(\mathbf{z}_t)}{\partial z_a}\sum_k \frac{G_{ak}(\mathbf{z}_t)\sqrt{\eta_k}}{2}\operatorname{tr}[(\hat{c}_k+\hat{c}_k^\dagger)\rho_t]\rho_t\,\mathrm{d}t \\
&+ \frac{\partial f(\mathbf{z}_t)}{\partial z_a}\sum_k \frac{G_{ak}(\mathbf{z}_t)}{2}\rho_t\mathrm{d}W_k \\
&+ \frac{1}{2}\frac{\partial^2 f(\mathbf{z}_t)}{\partial z_a\partial z_b}\sum_k \frac{G_{ak}(\mathbf{z}_t)G_{bk}(\mathbf{z}_t)}{4}\rho_t\,\mathrm{d}t \quad \texttt{[Itô correction from } \mathrm{d}f(\mathbf{z}_t)\texttt{]} \\
&+ \frac{\partial f(\mathbf{z}_t)}{\partial z_a}\sum_k \sqrt{\eta_k}\,\mathcal{M}[\hat{c}_k](\rho)\frac{G_{ak}(\mathbf{z}_t)}{2}\mathrm{d}t \quad \texttt{[Itô correction from } \mathrm{d}\rho_t\mathrm{d}f(\mathbf{z}_t)\texttt{]},
\end{aligned}
\tag{15}
$$

where have used Einstein's convention of summation on repeated indices for $a, b$. In this expression, the non-linear terms in $\rho_t$ cancel, and taking the average value $\mathbb{E}$ removes the Wiener processes. This gives

$$
\begin{aligned}
\frac{\mathrm{d}}{\mathrm{d}t}\mathbb{E}\big[\rho_t f(\mathbf{z}_t)\big] = \mathbb{E}\bigg\{ &-i f(\mathbf{z}_t)[H,\rho_t] + f(\mathbf{z}_t)\sum_{k=1}^{n}\mathcal{D}[\hat{c}_k](\rho_t)\,\mathrm{d}t \\
&+ \frac{\partial f(\mathbf{z}_t)}{\partial z_a}\bigg[F_a(\mathbf{z}_t)\rho_t + \sum_k \frac{\sqrt{\eta_k}\,G_{ak}(\mathbf{z}_t)}{2}(\hat{c}_k\rho_t + \rho_t\hat{c}_k^\dagger)\bigg] \\
&+ \frac{1}{2}\frac{\partial^2 f(\mathbf{z}_t)}{\partial z_a\partial z_b}\sum_k \frac{G_{ak}(\mathbf{z}_t)G_{bk}(\mathbf{z}_t)}{4}\rho_t\bigg\}.
\end{aligned}
\tag{16}
$$

We now use equation (14) defining $\rho_t(\mathbf{z})$ to convert the expectation values into integrals over $\mathbf{z}$:

$$
\begin{aligned}
\int f(\mathbf{z})\frac{\partial \rho_t(\mathbf{z})}{\partial t}\,\mathrm{d}^d\mathbf{z} = \int \mathrm{d}^d\mathbf{z}\bigg\{ &-i f(\mathbf{z})[H(\mathbf{z}),\rho_t(\mathbf{z})] + f(\mathbf{z})\sum_{k=1}^{n}\mathcal{D}[\hat{c}_k](\rho_t(\mathbf{z})) \\
&+ \frac{\partial f(\mathbf{z})}{\partial z_a}\bigg[F_a(\mathbf{z})\rho_t(\mathbf{z}) + \sum_k \frac{\sqrt{\eta_k}\,G_{ak}(\mathbf{z})}{2}(\hat{c}_k\rho_t(\mathbf{z}) + \rho_t(\mathbf{z})\hat{c}_k^\dagger)\bigg] \\
&+ \frac{1}{2}\frac{\partial^2 f(\mathbf{z})}{\partial z_a\partial z_b}\sum_k \frac{G_{ak}(\mathbf{z})G_{bk}(\mathbf{z})}{4}\rho_t(\mathbf{z})\bigg\}.
\end{aligned}
\tag{17}
$$

Finally, integrating by parts to bring the derivatives from $f$ to $\rho$, and noting that the resulting integral equation is true for all $f$ yields

$$
\begin{aligned}
\frac{\partial \rho_t(\mathbf{z})}{\partial t} = &-i[H(\mathbf{z}),\rho_t(\mathbf{z})] + \sum_{k=1}^{n}\mathcal{D}[\hat{c}_k](\rho_t(\mathbf{z})) \\
&- \frac{\partial}{\partial z_a}\bigg[F_a(\mathbf{z})\rho_t(\mathbf{z}) + \sum_k \frac{\sqrt{\eta_k}\,G_{ak}(\mathbf{z})}{2}(\hat{c}_k\rho_t(\mathbf{z}) + \rho_t(\mathbf{z})\hat{c}_k^\dagger)\bigg] \\
&+ \frac{1}{2}\frac{\partial^2}{\partial z_a\partial z_b}\bigg[\sum_k \frac{G_{ak}(\mathbf{z})G_{bk}(\mathbf{z})}{4}\rho_t(\mathbf{z})\bigg].
\end{aligned}
\tag{18}
$$

---

[3]This is the strategy that was followed in [29] to get a similar equation for the so called *open quantum Brownian motion* which is in some way the simplest continuous quantum–classical system one can think of (the classical variable is just the integrated signal).

This partial differential equation (18) is mathematically equivalent, by construction, to the direct stochastic representation of measurement (1) and feedback (12).

The present parameterization makes the interpretation of each term transparent. In addition to the intrinsic quantum dynamics with measurement-induced decoherence, we have a deterministic drift of the classical variables with one part that is intrinsic, given by $F_a(\mathbf{z})$ and one part that is of quantum origin, given by $\sqrt{\eta_k}\, G_{ak}(\mathbf{z})\,(\hat{c}_k\rho_t(\mathbf{z})+\rho_t(\mathbf{z})\hat{c}_k^\dagger)/2$. Finally, there is diffusion, positive by construction, that depends quadratically on $G$ which says how strongly the signal drives the classical dynamics.

### 3.3 Non-diagonal form and generality

To get closer to the notation used in [6], we need to break the "diagonality" of our equations and expand the $\hat{c}_k$ in a family of other operators $\hat{c}_k = \Gamma_{\alpha k}\hat{L}_\alpha$ where $\Gamma$ is a generic complex rectangular matrix and the $\hat{L}_\alpha$ are generic operators.

With this expansion, the SME for the density matrix can be rewritten as

$$
\begin{aligned}
\mathrm{d}\rho =&-i[H,\rho]\,\mathrm{d}t + \sum_k \left\{ \Gamma_{\alpha k}\Gamma_{\beta k}^* \left( \hat{L}_\alpha \rho \hat{L}_\beta^\dagger - \frac{1}{2}\{\hat{L}_\beta^\dagger \hat{L}_\alpha,\rho\} \right)\mathrm{d}t \right. \\
&\left. + \sqrt{\eta_k}\,\Gamma_{\alpha k}\left(\hat{L}_\alpha - \langle \hat{L}_\alpha\rangle\right)\rho\,\mathrm{d}W_k + \sqrt{\eta_k}\,\Gamma_{\alpha k}^*\rho\left(L_\alpha^\dagger - \langle \hat{L}_\alpha^\dagger\rangle\right)\mathrm{d}W_k \right\},
\end{aligned}
\tag{19}
$$

with $\langle L\rangle := \mathrm{tr}[L\rho]$. For the classical variables $z_a$ we obtain

$$
\mathrm{d}z_a = F_a\,\mathrm{d}t + \sum_k \frac{G_{ak}\sqrt{\eta_k}}{2}\Gamma_{\alpha k}\langle \hat{L}_\alpha\rangle\,\mathrm{d}t + \frac{G_{ak}\sqrt{\eta_k}}{2}\Gamma_{\alpha k}^*\langle \hat{L}_\alpha^\dagger\rangle\,\mathrm{d}t + \frac{G_{ak}}{2}\mathrm{d}W_k.
\tag{20}
$$

We may now explicitly relate our parameterization with that of [6] by introducing

$$
D_0^{\alpha\beta} = \sum_k \Gamma_{\alpha k}\Gamma_{\beta k}^* = (\Gamma\Gamma^\dagger)_{\alpha\beta} \succeq 0,
\tag{21}
$$

$$
\sigma_{ak} = \frac{G_{ak}}{2},
\tag{22}
$$

$$
D_1^{a\alpha} = \left(\sigma\sqrt{\eta}\,\Gamma^\dagger\right)_{a\alpha} = \sum_k \sigma_{a,k}\sqrt{\eta_k}\Gamma_{\alpha,k}^*.
\tag{23}
$$

This gives the SME for the quantum density matrix

$$
\begin{aligned}
\mathrm{d}\rho =&-i[H,\rho]\,\mathrm{d}t + D_0^{\alpha\beta}\left(\hat{L}_\alpha\rho\hat{L}_\beta^\dagger - \frac{1}{2}\{\hat{L}_\beta^\dagger\hat{L}_\alpha,\rho\}\right)\mathrm{d}t \\
&+ \sum_k(\sigma^{-1}D_1^*)_{k\alpha}\left(\hat{L}_\alpha - \langle\hat{L}_\alpha\rangle\right)\rho\,\mathrm{d}W_k + (\sigma^{-1}D_1^*)_{k\alpha}\rho\left(L_\alpha^\dagger - \langle\hat{L}_\alpha^\dagger\rangle\right)\mathrm{d}W_k,
\end{aligned}
\tag{24}
$$

with the generalized inverse where $\sigma^{-1} := \sigma^T(\sigma\sigma^T)^{-1}$. The classical dynamics is then simply

$$
\mathrm{d}z_a = F_a\,\mathrm{d}t + D_1^{*\,a\alpha}\langle\hat{L}_\alpha\rangle\,\mathrm{d}t + D_1^{a\alpha}\langle\hat{L}_\alpha^\dagger\rangle\,\mathrm{d}t + \sum_k\sigma_{ak}\,\mathrm{d}W_k.
\tag{25}
$$

The equations (24) and (25) correspond to the unraveled form of the most general dynamics presented in [6]. We may choose to parameterize the dynamics this way, as a function of $D_0$, $D_1$, and $\sigma$ but the dynamics is then consistent only if non-trivial positivity constraints are verified. For example we have

$$
D_1 D_0^{-1} D_1^\dagger = \frac{1}{4} G\sqrt{\eta}\,\Gamma^\dagger(\Gamma\Gamma^\dagger)^{-1}\Gamma\sqrt{\eta}\,G^T = \frac{G\eta G^T}{4},
\tag{26}
$$

hence, introducing $2D_2 = \sigma\sigma^T = \frac{GG^T}{4}$ we have $2D_2 \succeq D_1 D_0^{-1} D_1^{\dagger}$, which corresponds to the so called "decoherence diffusion trade off" [6]. Its saturation is straightforward to understand in the diagonal formalism: it corresponds to setting the measurement efficiencies $\eta_k$ at 1. In fact, works on Newtonian gravity were all done with $\eta_k = 1$, and were saturating the trade-off before it was put forward.

Finally, we can rewrite the PDE (18) in terms of the $D$ matrices we just introduced:

$$
\begin{aligned}
\frac{\partial \rho_t(\mathbf{z})}{\partial t} = {} & -i[H(\mathbf{z}),\rho] + D_0^{\alpha\beta}(\mathbf{z})\left(\hat{L}_\alpha \rho \hat{L}_\beta^{\dagger} - \frac{1}{2}\{\hat{L}_\beta^{\dagger}\hat{L}_\alpha, \rho\}\right) \\
& - \frac{\partial}{\partial z_a}\left[D_1^{0a}(\mathbf{z})\rho_t(\mathbf{z}) + (D_1^{a\alpha}\hat{L}_\alpha \rho_t(\mathbf{z}) + \rho_t(\mathbf{z})D_1^{a\alpha*}\hat{L}_\alpha^{\dagger})\right] \\
& + \frac{1}{2}\frac{\partial^2}{\partial z_a \partial z_b}\left[D_2^{ab}(\mathbf{z})\rho_t(\mathbf{z})\right],
\end{aligned}
\tag{27}
$$

where $D_1^{0a} = F_a$, to get as close as possible to the notations in [6]. This is the most general parameterization of a second order PDE in $\mathbf{z}$ preserving the properties of $\rho$. We could have started from it and then realized that ensuring the consistency of the resulting dynamics forces the non-trivial positivity constraint $2D_2 \succeq D_1 D_0^{-1} D_1^{\dagger}$.

Emphasizing the glue linking quantum and classical sectors, namely the measurement signal, we thus obtained a parameterization that is interpretable, consistent by construction (the parameters themselves are not further constrained by equations), but no less general.

## 4 The Markovian feedback limit

The dynamics we have considered is already Markovian in $\mathbf{z}$ and $\rho$. However, if the intrinsic dynamics of the classical system is much faster than the quantum dynamics, we enter a regime that corresponds to Markovian (measurement based) *feedback* [10,30,31]. The equations then drastically simplify, and the empirical predictions can be computed from a simple Lindblad equation acting on quantum degrees of freedom only. Historically, the hybrid models used for gravity immediately took the Markovian feedback limit [1–3] which was relevant to the Newtonian setting. While it allowed to get quantitative predictions in this relevant regime, it unfortunately obfuscated the link with the general case. Here, we can simply see the Markovian feedback limit as a special case.

Deriving the Markovian limit from the dynamics of $\mathbf{z}$ is in general problem dependent, and can be non-trivial if $\mathbf{z}$ has non-linear dynamics (see *e.g.* [32]). However, it is possible to write *all* the possible dynamics that one can ultimately get this way. In this endeavor, the measurement signal again plays the central role.

Concretely, the Markovian limit requires that the backaction the classical variables have on the quantum part is memoriless, *i.e.* that it depends only on the instantaneous signal. The latter has the regularity of white noise, and thus can enter only linearly in the dynamics, which means that the backaction can appear only in the potential part $H(\mathbf{z}) = H_0 + V(\mathbf{z})$ with

$$
V(\mathbf{z})\,\mathrm{d}t = \sum_k \hat{b}_k \,\mathrm{d}r_k,
\tag{28}
$$

where the $\hat{b}_k$ are Hermitian operators. The fluctuating potential given by equation (28) has to be interpreted with care, and is notoriously subtle![4] The corresponding unitary acts infinitesimally *after* the signal is sourced from the measurement dynamics (1), and thus we should

---

[4]Crucially, it is *not* sufficient to just interpret (28) in Stratonovich form, which gives an incorrect operator ordering [33].

understand (28) to mean

$$\rho_t + d\rho_t = e^{-i\sum_k \hat{b}_k dr_k}\left(\rho_t + d\rho_t^{(\text{meas})}\right)e^{+i\sum_k \hat{b}_k dr_k}. \tag{29}$$

where $d\rho^{(\text{meas})}$ is given by (1). While this form is intuitively correct, and convenient to carry further derivations, it can be further motivated by smoothing the signal at a timescale $\varepsilon$ and sending $\varepsilon$ to zero (see *e.g.* [10]).

Expanding the right-hand side of (28) and using the Itô rule $dW_k dW_\ell = \delta_{k\ell} dt$ gives

$$d\rho_t = -i[H_0, \rho_t]dt + \sum_k \left\{ \mathcal{D}[\hat{c}_k](\rho_t)dt + \mathcal{M}[\hat{c}_k](\rho_t)dW_k \quad \texttt{[measurement]} \tag{30}\right.$$

$$-i\left[\hat{b}_k, \rho_t\right]\left(\frac{1}{2\sqrt{\eta_k}}dW_k + \frac{1}{2}\text{tr}[(\hat{c}_k + \hat{c}_k^\dagger)\rho_t]dt\right) \quad \texttt{[standard feedback]} \tag{31}$$

$$-\frac{1}{8\eta_k}\left\{\hat{b}_k^2, \rho_t\right\}dt \quad \texttt{[Itô correction from each } e^{\pm i\sum_k \hat{b}_k dr_k}\texttt{]} \tag{32}$$

$$+\frac{1}{4\eta_k}\hat{b}_k\rho\hat{b}_k dt \quad \texttt{[Itô correction from } e^{\pm i\sum_k \hat{b}_k dr_k}\texttt{cross terms]} \tag{33}$$

$$\left.-\frac{i}{2\sqrt{\eta_k}}\left[\hat{b}_k, \mathcal{M}[\hat{c}_k](\rho_t)\right]dt \quad \texttt{[Itô correction from } e^{\pm i\sum_k \hat{b}_k dr_k}\texttt{/meas]}\right\}. \tag{34}$$

The non-linear term in the standard (or naively expected) feedback part (31) cancels with the non-linear part of the Itô correction (34), and the Itô corrections (32) and (33) combine into a new Lindblad term. With these simplifications one obtains

$$d\rho_t = -i[H_0, \rho_t]dt + \sum_k \mathcal{D}[\hat{c}_k](\rho_t)dt + \mathcal{M}[\hat{c}_k](\rho_t)dW_k$$
$$+ \sum_k \frac{-i}{2\sqrt{\eta_k}}\left[\hat{b}_k, \hat{c}_k\rho_t + \rho_t\hat{c}_k^\dagger\right]dt + \frac{1}{4\eta_k}\mathcal{D}[\hat{b}_k](\rho_t) - i\left[\hat{b}_k, \rho_t\right]\frac{1}{2\sqrt{\eta_k}}dW_k. \tag{35}$$

In this Markovian limit, the classical variables no longer have dynamics of their own and are slaved to the quantum ones via the signal equation (4). Conceptually, the stochastic dynamics is still important. If this hybrid dynamics is produced effectively in a lab with measurement and feedback, then the stochastic trajectory can be reconstructed from the measured signal, and thus we would lose important information by averaging over it. If the hybrid dynamics is thought to be fundamental, then the signal is not directly observable, but the stochastic dynamics yields a progressive collapse of the quantum state that is useful for foundational purposes (*e.g.* to solve the measurement problem).

If we are interested only in empirical predictions in the hybrid-dynamics-as-fundamental-model case, then we cannot make better predictions with the stochastic description than with its average $\bar{\rho} := \mathbb{E}[\rho]$ where $\mathbb{E}$ is the stochastic average over the signal. In this Markovian setup, this gives a simple Lindblad equation

$$\frac{d}{dt}\bar{\rho}_t = -i[H_0, \bar{\rho}_t] + \sum_k \frac{-i}{2\sqrt{\eta_k}}\left[\hat{b}_k, \hat{c}_k\bar{\rho}_t + \bar{\rho}_t\hat{c}_k^\dagger\right] + \mathcal{D}[\hat{c}_k](\bar{\rho}_t) + \frac{1}{4\eta_k}\mathcal{D}[\hat{b}_k](\bar{\rho}_t). \tag{36}$$

That equation separates neatly the measurement-induced decoherence and the decoherence induced by the randomness in the feedback. However, because of the peculiar form of the second term, the Lindblad structure is not obvious and the unitary contribution from the feedback is unclear. It is possible to make things more transparent by noting that:

$$\sum_k \frac{-i}{2\sqrt{\eta_k}}\left[\hat{b}_k, \hat{c}_k\bar{\rho} + \bar{\rho}\hat{c}_k^\dagger\right] + \mathcal{D}[\hat{c}_k](\bar{\rho}) + \frac{1}{4\eta_k}\mathcal{D}[\hat{b}_k](\rho)$$
$$= \sum_k \frac{-i}{4\sqrt{\eta_k}}\left[\hat{b}_k\hat{c}_k + \hat{c}_k^\dagger\hat{b}_k, \bar{\rho}\right] + \mathcal{D}\left[\frac{\hat{b}_k}{2\sqrt{\eta_k}} + i\hat{c}_k\right](\bar{\rho}). \tag{37}$$

This forms makes it natural to introduce the effective potential $V^{\text{eff}} := \sum_k \frac{1}{4\sqrt{\eta_k}} \left( \hat{b}_k \hat{c}_k + \hat{c}_k^\dagger \hat{b}_k \right)$, which gives the master equation its alternative form

$$\frac{\mathrm{d}}{\mathrm{d}t} \bar{\rho}_t = -i \left[ H_0 + V^{\text{eff}}, \bar{\rho}_t \right] + \sum_k \mathcal{D} \left[ \frac{\hat{b}_k}{2\sqrt{\eta_k}} + i \hat{c}_k \right] (\bar{\rho}_t). \tag{38}$$

*Remark* 7 (Unitary potential and disentanglement). If the Hilbert space of the system can decomposed into a tensor product of factors (for example, into factors associated to different distinguishable particles, or different regions of space), and if each $\hat{c}_k$ acts on a single factor, the pure measurement dynamics given by the SME (1) manifestly preserves product states. Then, if the feedback operators $\hat{b}_k$ are also each acting on a single factor, the whole measurement and feedback dynamics preserves product states at the stochastic level. However, the potential $V_{\text{eff}}$ taken alone is generically entangling if $\hat{c}_k$ and $\hat{b}_k$ act on different factors. Measurement and feedback introduce precisely the right amount of decoherence / noise to kill the entanglement created by the effective potential. Provided one accepts extra decoherence and noise, any unitary potential $V_{\text{eff}}$ can be reproduced through measurement and feedback [34], and thus simulated in an efficient way classically by evolving a stochastic pure state.

*Remark* 8 (Principle of least decoherence). In the Markovian feedback limit, the diffusion on the classical variables becomes pure decoherence on the quantum part. Hence, we now have two sources of decoherence, one from the measurement and one from the feedback. If we fix the strength of the effective potential, which is proportional to the product $\hat{c}_k \hat{b}_k$, then decreasing one source of decoherence mechanically increases the other. Intuitively, lowering the decoherence coming from the measurement part increases the diffusion of the classical variables which, via the feedback potential, increases the noise on the quantum part. This ultimately gives decoherence at the master equation level. The decoherence–diffusion trade-off consequently becomes a decoherence–decoherence trade-off, and thus the total decoherence is *lower bounded*. Asking that a model be at this decoherence minimum seems like a physically natural way to fix its parameters, and this is what was called the principle of least decoherence in [35]. Note that the existence of this lower bound is a crucial difference between hybrid quantum–classical dynamics and simple collapse models (corresponding to measurement without feedback): one cannot escape experimental falsification by reducing decoherence arbitrarily. Provided one falsifies the model at the least decoherence point, one falsifies all models reproducing a given effective potential.

*Remark* 9 (Dissipation from pure measurement and feedback). From the Lindblad equation in the form of (38), we observe an interesting property.[5] It is standard to take $\hat{c}_k = \hat{c}_k^\dagger$, which gives "pure" (or non-demolition, non-dissipative continuous measurement,[6]) which corresponds to the intuitive notion of quantum measurement. More precisely, if we measure a single Hermitian $\hat{c}$, the stochastic evolution (1) progressively sends $\rho$ to one of the eigenstates of $\hat{c}$ with a probability given by the Born rule. Moving away from self-adjoint measurement operators is equivalent to adding dissipation. From equation (38), we see that even if we start from a pure measurement in that sense, we can tune the feedback operators $\hat{b}_k$ to obtain *any* non-Hermitian operator $\frac{\hat{b}_k}{2\sqrt{\eta_k}} + i \hat{c}_k$ in $\mathcal{D}$. Then, upon redefining $H_0$ to absorb the unitary contribution of the feedback, we can obtain *all* possible *dissipative* Lindblad dynamics. This is important because this means that even if the pure measurement part is without dissipation, driving the quantum part to infinite temperature, feedback can cool the stationary state down.

---

[5]To my knowledge, this observation was first made by Lajos Diósi.

[6]Note the slightly misleading terminology used in open quantum systems: a term of the form $\mathcal{D}[\hat{c}](\rho)$ appearing in the Lindblad equation is called a *dissipator* but it does not induce dissipation in the usual sense if $\hat{c} = \hat{c}^\dagger$ (only decoherence).

The lessons derived in the Markovian feedback limit are important for more general hybrid quantum–classical models. First, because the latter often reduce to the Markovian feedback models in appropriate physical limits, and thus the results obtained in the Markovian limit can meaningfully constrain their parameters. Second, because most conclusions obtained in the Markovian feedback limit likely subsist at least qualitatively in the general case. For example, the fact that diffusion in the classical variables is empirically indistinguishable from decoherence on the quantum part is true only in the Markovian limit. But this suggests that in the general case, diffusion in the classical variable likely has *very similar* consequences to decoherence, and thus that too much diffusion can be falsified in just the same way.

## 5 Discussion

### 5.1 Modular construction of (candidate) fundamental theories

The general idea we have leveraged in this note is that quantum–classical dynamics can be decomposed into pure measurement, where the quantum only influences the classical, and a feedback part, where the classical back-reacts on the quantum. I believe this split clarifies the construction of candidate fundamental theories of Nature (even retrospectively for Oppenheim's approach), and is thus particularly useful for theory builders.

In the context of fundamental theories, the feedback part of the dynamics is usually well constrained or even already tested. For gravity, we know how quantum matter evolves in a fixed gravitational background. The answer, quantum field theory in curved space-time, is not without technical difficulties, but is well tested at least in some limits (*e.g.* by dropping an atom in the Earth gravitational field). As far as I know, no one doubts its validity, at least in the fixed background context. The difficulty lies in sourcing a classical gravitational field from quantum matter. For this part of the dynamics, there is currently no experimental guide available, *even in the Newtonian limit* (since we do not even know if gravity is quantum, hybrid, or something else entirely). In our construction of hybrid dynamics, this tricky open part, the "sourcing", is done by the continuous measurement process, which extracts a classical signal from quantum degrees of freedom.

For the theory builder, the main choice thus lies in picking the appropriate measured operators $\hat{c}_k$. In the non-relativistic limit of gravity, it should intuitively be something related to the mass density of quantum matter at every point $\propto \hat{M}(\mathbf{x})$, or directly to the gravitational field this matter density produces $\propto \hat{\Phi}(\mathbf{x}) = -\int \mathrm{d}^3\mathbf{y} \frac{\hat{M}(\mathbf{y})}{|\mathbf{x}-\mathbf{y}|}$, two options considered in [3]. Even in the non-relativistic context, this choice yields an infinite amount of decoherence, which can be regulated by smearing the mass density over a cut-off distance $\sigma$. Fortunately, this is fine for gravity, which has not been tested at very short distances. With this regulator, the former choice $\hat{M}(\mathbf{x})$ yields a measurement part of the dynamics identical to that of the continuous spontaneous localization model (CSL) [36], while the latter choice, $\hat{\Phi}(\mathbf{x})$, yields the Diósi-Penrose model [37,38]. The feedback part is Markovian, and thus the complete models, with the backaction of gravity on matter, are obtained along the lines of section 4.

For these non-relativistic models, there exists a range of continuous measurement rates that is not yet falsified and compatible with (non-relativistic) observations.[7] However, as argued in remark 8, the models are in principle falsifiable for all values of their parameters. Interestingly, asking for the model with the least amount of extra decoherence singles out the measured operator $\hat{\Phi}(\mathbf{x})$, and thus gives exactly the decoherence of the Diósi-Penrose model, which was

---

[7]In the SME (1), we are free to multiply the measured operator by a constant, to increase or reduce the strength of the continuous measurement. The dimension of this constant depends on the operator being measured, but is proportional to the root of a rate.

only motivated heuristically before [35]. In any case, the existence of such non-trivial models demonstrates that hybrid quantum–classical gravity need not yield ridiculous paradoxes or gross experimental violations [4, 39]. Of course, this successful non-relativistic sanity check is not sufficient to demonstrate that hybrid gravity models are viable.

## 5.2 The (relativistic) measurement part: Collapse phenomenology and divergences

The main merit of Oppenheim's program [11], further refined and expanded with Layton, Šoda, Sparaciari, and Weller-Davies [6, 28, 32, 40–43], has been to push hybrid dynamics beyond the non-relativistic playground where it had remained confined. More than a model, these works build and refine a framework, from which one can in principle specify a variety of models. In our language, the choice of measurement operators $\hat{c}_k$ and feedback dynamics is not fully explicit. Rather, a set of non-trivial requirements are derived that would make the resulting dynamics properly relativistic and have General Relativity as classical limit. It is not yet known if there exists choices that satisfy all the requirements jointly and are compatible with observations.

In my opinion, the measurement part of the dynamics is the most important and difficult to make consistent with observations for a relativistic theory.[8] Indeed, constructing consistent relativistic continuous measurement equations (equivalently relativistic collapse models), which is required in Oppenheim's program as in every other attempt at making hybrid dynamics relativistic, is notoriously difficult. The main hurdle is that the standard approach [45] with local dissipators $\mathcal{D}[\mathcal{O}(x)]$ generically induces an infinite spontaneous heating of the vacuum. One could think of renormalizing this effect, and for various models this can be done formally [46]. However, the required counter terms are negative and thus break the Lindblad form of the master equation, which is unacceptable for a fundamental theory.

It is tempting, as a fallback option, to just *regularize* the dynamics, without renormalizing with counter terms. But then one should keep in mind that present constraints on non-relativistic collapse models make the resulting scale dangerously large (far larger than the Planck scale, and even larger than the nucleon scale [47–49]).

An even bolder alternative is to relax the Markovianity of the measurement part itself, to covariantly smear the dissipators in space-time. One then enters the realm of so called non-Markovian unraveling of open systems [50–52] or non-Markovian collapse models [53]. Such models have empirical consequences that are essentially as mild as one desires. While I once thought this was a promising route, I now think it comes with too much flexibility, as it seems to allow hiding any fully quantum theory into a collapse equation [54, 55]. Further, it is known that the resulting stochastic trajectories do not have a continuous measurement interpretation [56, 57]. As a result, there is no obvious equivalent of the signal that one can use as quantum–classical glue, and thus no clear route towards consistent hybrid dynamics.

Since the relativistic continuous measurement part is currently difficult to make consistent on its own after about 35 years of efforts, one may hope that the solution can be found *only* by turning on the feedback, and considering the backaction of the classical sector. While the total amount of decoherence cannot be decreased this way, the asymptotic heating in principle can. Indeed, from the Markovian feedback limit, we saw in remark 9 that a well chosen feedback can induce dissipation (and thus in principle cooling). In hybrid models of Newtonian gravity, the measurement and feedback operators commute [3], and thus no dissipation appears from the feedback; but this could in principle be an effect showing up only in higher post-Newtonian orders. However, without further evidence, expectations should remain moderate.

---

[8]A similar opinion has been defended independently by Diósi in a recent note [44].

### 5.3 The relativistic feedback part: Classical stochastic dynamics

As the measurement part of relativistic dynamics remains an open problem, it is tempting to temporarily put it aside and explore pure feedback dynamics. In our language, this means considering dynamics where the measurement operators $\hat{c}_k$ are fixed to zero and thus where the signal (4) becomes pure noise. The dynamics is then no longer really "hybrid", in the sense that we obtain a theory of quantum matter moving in a stochastic background sourced independently of the matter distribution. The freedom only lies in fixing how this pure noise enters in the classical equations of motion (12).

Recent works by Grudka, Oppenheim, Russo, and Sajjad [58] and Russo and Oppenheim [59] have discussed promising properties of particular stochastic theories of gravity that can be constructed this way. Assuming some subtle issues of positivity can be addressed (the kernel $D_2$ they use in their non-diagonal formulation is, strictly speaking, not positive), they find in [58] that the models can be made asymptotically safe, *i.e.* renormalizable in a strong sense.

These works are intriguing, and may well revive the interest in stochastic models of pure gravity. However, because the problematic continuous measurement part is left unaddressed, and because this is where most of the identified difficulties of the hybrid program lie, the implications for the construction of a consistent model of classical gravity coupled to quantum matter are at the very least unclear. In particular, the favorable renormalizability properties of a noisy classical sector do not imply anything for the complete hybrid model, since, again, it is on the quantum part (continuous measurement) that problems have been identified.

### 5.4 Conclusion

We have reconstructed the most general continuous hybrid quantum–classical dynamics from the theory of measurement and feedback. This has allowed us to relate various mathematical formulations of the dynamics, clarify its subtle Markovian limit, and, most importantly, benefit from a powerful physical intuition pump.

Ultimately, we do not yet know if hybrid quantum–classical dynamics play an important role in the fundamental laws of our universe. This is certainly an option worth considering, and it is important to try to address the theoretical difficulties previously mentioned. Meanwhile, we should remember that quantum–classical dynamics are routinely realized as effective descriptions in the laboratory. This continuous measurement and feedback realization is a way to re-derive the most general equations but also, currently, their main application.

## Acknowledgments

I thank Lajos Diósi for our collaboration which helped shape my thoughts on these questions, and Pierre Guilmin for keeping my interest in them alive. I also thank Zach Weller-Davies for helpful comments and discussions. I am finally grateful for the constructive comments made by the three anonymous referees.

**Funding information** This work was funded in part by the European Union (ERC, QFT.zip project, Grant Agreement no. 101040260). Views and opinions expressed are however those of the author(s) only and do not necessarily reflect those of the European Union or the European Research Council Executive Agency. Neither the European Union nor the granting authority can be held responsible for them.

# A   Itô rules for busy physicists

Itô calculus is the mathematical machinery that allows to work with multiplicative white noise rigorously and without ambiguity. It is well explained in many textbooks, like that of Øksendal [60], and I only summarize here non rigorously what is needed to study hybrid dynamics.

The difficulty in writing stochastic differential equations driven by white noise, *i.e.* the derivative of a Brownian motion, is that the latter is not differentiable in the usual sense. Writing the Brownian motion $W$, $\frac{dW}{dt}$ is not a well defined function (but it is a distribution). The idea of Itô calculus is to define the differentiation for such processes implicitly by defining instead an integral against white noise:

$$ ``\int_0^T f(t)\frac{dW}{dt}\,dt\text{''} := \int_0^T f(t)\,dW\,. \tag{A.1}$$

While the left-hand side integral integral is a priori problematic, the right-hand side can be defined rigorously with the Itô integral.

To define the Itô integral, one simply constructs it as the limit of a Riemann sum:

$$\int_0^T f(t)\,dW := \lim_{N\to+\infty}\sum_{k=1}^{N-1} f(k/N)[W((k+1)/N)-W(k/N)]\,. \tag{A.2}$$

Crucially, if $f$ itself depends on $W$, and unlike with the standard Riemann integral, the specific choice of Riemann sum matters. For example, replacing $f(k/N)$ with the symmetric $[f(k/N)+f((k+1)/N)]/2$, gives a different stochastic integral, the Stratonovich integral.

Once the Itô integral is defined, and as a pure notational convenience, we may drop the integral sign and write $f(t)\,dW$. This is really just what we would like intuitively, namely $f(t)\frac{dW}{dt}$, except multiplied by $dt$. The previous definition generalizes from $W$ to any stochastic process with the same (or better) regularity.

Importantly, with this choice of Riemann sum, the Itô integral has zero expectation value

$$\mathbb{E}\left[\int_0^T f(t)\,dW\right] = 0\,, \tag{A.3}$$

which is extremely convenient to go from the full stochastic description of continuous measurement dynamics to its averaged Lindblad representation. However, standard differentiation rules need to be replaced with the Itô rule. Heuristically, the need for a modification comes because $dW$ is of order $\sqrt{dt}$ and thus when Taylor expanding a function, we need to go to second order in $dW$ to get the order $dt$ right. For a function $g$ of a scalar stochastic process $X_t$ obeying the SDE $dX_t = a(t)dt + b(t)dW$, we have Itô's lemma:

$$df(X_t) = \frac{\partial f}{\partial X}\,dX_t + \frac{b(t)^2}{2}\frac{\partial^2 f}{\partial X^2}\,dt\,, \tag{A.4}$$

which is easily derived from the definition (A.2). Itô's lemma is equivalent to the simple "physicist" Itô rule, which consists in expanding everything to second order in $dW$ and using $dW\,dW = dt$. For example, consider a single Brownian motion $W$ and the function $W^2$

$$d(W^2) = dW\,W + W\,dW + dW\,dW = 2W\,dW + dt\,, \tag{A.5}$$

which coincides with what one would have obtained from Itô's lemma. This rule generalizes naturally to several independent Brownian motions into

$$dW_k\,dW_\ell = \delta_{k\ell}\,dt\,. \tag{A.6}$$

Practically, this latter formulation of the rule is convenient when $X_t$ is matrix valued, and not commuting with its differential, a situation in which applying Itô's lemma in its standard form can be tedious. For example, we could easily compute the Itô derivative of $\rho^2$ where $\rho$ obeys the continuous measurement SME (1) using the physicist rule:

$$\mathrm{d}(\rho^2) = \mathrm{d}\rho\,\rho + \rho\,\mathrm{d}\rho + \mathrm{d}\rho\,\mathrm{d}\rho \tag{A.7}$$

$$= -i\{[H_0, \rho], \rho\}\,\mathrm{d}t + \sum_{k=1}^n \{\mathcal{D}[\hat{c}_k](\rho), \rho\}\,\mathrm{d}t + \sqrt{\eta_k}\{\mathcal{M}[\hat{c}_k](\rho), \rho\}\,\mathrm{d}W_k, + \eta_k^2\,\mathcal{M}[\hat{c}_k]^2(\rho)\,\mathrm{d}t\,. \tag{A.8}$$

This Itô rule (A.6) and the fact that Itô integrals against the Wiener process are zero (A.3) is about all that one needs for continuous measurement.

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
