# Peer review of "General quantum-classical dynamics as measurement based feedback"

_SciPost Physics, doi:SciPost Phys. 17, 083 (2024)_

## Round 2 · Referee Report · Anonymous (Referee 1) · 2024-6-3

Strengths

1. Very well written.
2. Provides a synergetic link between different research areas.
3. Reviews continuous measurement and feedback well.

Weaknesses

1. Does not review in sufficient detail the recent work of Oppenheim and co-workers, and its relation to previous work on semiclassical gravity.

Report

I think this paper serves a very useful purpose to bridge the divide between: (a) some recent, very prominent, claims about the promise of a supposedly new approach to semiclassical general relativity; (b) a body of literature on semiclassical gravity in the Newtonian regime which is already well linked to work on spontaneous collapse; and (c) a large body of literature on continuous measurement and feedback. Therefore I think it could be valuable enough to publish in SciPost. However, I think the author should devote more space to explaining the research claims and results in these recent papers by Oppenheim and co-workers. Many of the potential readers of this paper may not be familiar with that work, beyond the popular discourse on it. What is novel in those works compared to the Newtonian-regime papers, and what are the short-comings (see also requested change 9 below)?

Requested changes

In addition to the major weakness discussed above, I have these requests for changes:

1. Equation (1) is only one model of continuous measurement -- the other is a jump unravelling of a master equation. This is eventually mentioned, but only 2 pages later. It would be better not to hold the knowledgeable reader is suspense for so long.

2. Equation (1) uses the superoperator notation of the textbook [1] for the deterministic part but not for the stochastic part. Is there any particular reason to replace H with M?

3. It is claimed (correctly) that Eq. (9) is the most general form of feedback, but it is not explained why this is so.

4. The statement “but all objects are now implicitly z dependent” may be misleading. It is exactly the 3 objects (or sets of objects) mentioned in the preceding sentence that can have z-dependent.

5. The paper needs a hyphens in compound adjectives, e.g. in z-dependent, measurement-induced etc. It also needs an n-dash in the adjectival phrase "hybrid quantum--classical".

6. The phrase "the “diagonal” form" is used on page 4 a few paragraphs before it is explained what this means.

7. Below Eq.(32), what follows “noting that” is well-known in previous literature. If the author can find where this was first noted, at least in this context, and reference it, that would be of service to the community.

8. Regarding “Further, since the continuous measurement interpretation disappears, it is unclear if such models can be used as a basis for consistent hybrid dynamics”, could the author say something stronger here? Or at least expand upon the problem for those unfamiliar with it.

9. With regard to the major issue mentioned in the report, here is an example of where more information would be highly desirable: “However, because the problematic quantum measurement part is left unaddressed ...” Why and how was this left unaddressed by Oppenheim and co-workers?

Recommendation

Ask for major revision

---

## Round 2 · Referee Report · Anonymous (Referee 2) · 2024-6-9

Strengths

1. It provides a clear and mathematically consistent connection between the dynamics of quantum systems under continuous measurement and feedback, and the recent results on the form of the more general hybrid quantum-classical dynamics (Ref 5).

2. The treatment exploits the intuition provided by the known results on the dynamics of measured quantum systems in order to clarify the typical features of a generic hybrid classical-quantum dynamics and the related decoherence and diffusion effects.

3. The paper, adopting its "diagonal form" , has the merit that complete positivity of the dynamics is naturally obtained and one has a simplified form of the general approach of Ref. [5].

4. The conclusions, even though somehow very general, provide a broad picture of the challenges one has to afford in trying to derive fundamental hybrid quantum-classical theories.

5. The paper provides a quite direct connection between hybrid models and collapse models, which in some approaches does not seem so obvious.

Weaknesses

1. The alternative approach in which jump stochastic processes (such as photodetection) replace weak continuous measurements (i.e. homodyne/heterodyne like measurements, with a strong local oscillator) is mentioned only briefly. It is not evident if this at the end does not change too much the physics or not. A more explicit comment on what can be achieved considering jump processes and eventually a more explicit comparison with Refs. 10 and 22 could be extremely useful to the reader.

2. The derivation of the Markovian limit is a bit misleading in my opinion (see below)

Report

In my opinion the paper deserves publication in SciPost, as it perfectly fulfills its scope, that is, "Provide a novel and synergetic link between different research areas.". Such a synergetic link can be really fruitful for both fields, as already pointed out in the list of strengths provided above. In my opinion the paper only deserves to clarify few points in order to become readable by a broader set of scientists.

Requested changes

1. The alternative approach based on jump stochastic processes should be discussed in more detail and not only almost incidentally as it is done in the present version. I understand that the mathematics can be different and one cannot exploit the effectiveness of the Ito calculus, but a more detailed discussion of the approach of Ref. 10 and 22 would provide a broader and clearer view.

2. The derivation of Sec IV of the Markovian limit in which the dynamics of the classical variables is adiabatically eliminated should be adjusted in my opinion. In fact, The equations, starting from eq. 24 seem to be written only for a single feedback term, i.e., a single feedback operator b_k. Instead I expect that, in order to be more general, we have many operators b_k, one for each measured signal r_k. However in most of the equations below the terms with b_k are always outside the sum over the index k.
This makes the derivation misleading and I think it would be proper to adjust it, since the beginning, by assuming a general sum over k in the fluctuating potential of Eq. (24).

Recommendation

Ask for minor revision

---

## Round 2 · Referee Report · Anonymous (Referee 3) · 2024-6-13

Strengths

1. Clarity of writing.

2. Handling stochastic master equations in a very friendly way for physicists, yet making statements that could be made rigorous.

3. Clearly drawn connection to known results in quantum optics that help in linking different communities.

Weaknesses

1. Connection to the literature, both in view of the possible physical applications and in view of the general formalism, is not adequate.

2. The focus of the paper should be more clearly stated.

3. In the presentation it is not always clear what is taken for granted and what is derived.

Report

The paper provides in a very readable way the connection between the stochastic master equation for continuous measurement with diffusive noise and the hybrid classical-quantum Fokker-Planck equations considered in Ref.[5]. The idea is to start from the stochastic master equation for continuous measurement and use the measured signal as noise source for the stochastic evolution of classical variables, on which the Hamiltonian and the measuring operators can be made dependent. The connection is clearly outlined, so that I find the paper deserves publication in that it provides useful links between different formulations. I find however that a better discussion of the potential physics that could be described and that motivated e.g. Ref.[5] would be in place. Indeed the formalism itself has been discussed also in previous publications, e.g. the quoted Ref.[22], where also jump noise has been considered. The neglect of jump measurements should be clarified. Combining the two kinds of noise does not necessarily lead to a trivially additive effect.

Requested changes

A better introduction to the addressed physical background.

Clearer motivation for not dealing with jump noise and the possible new implications in considering it.

The abstract mentions considering non-Markovian feedback, but apparently, the point is not further discussed.

The wording "without additional positivity constraints" in the abstract is somehow deceitful to me. The stochastic differential equation and the corresponding Fokker-Planck are equivalent.

Summations over k appear and disappear and should be made more consistent in notation to avoid confusing the reader.

Recommendation

Ask for minor revision

---

## Round 3 · Referee Report · Anonymous (Referee 1) · 2024-8-15

Report

The author has considerably improved the accessibility and clarity of the paper by his changes. I think it will be read with interest by many physicists interested in the state of the art of hybrid quantum-classical gravity speculation. I have no hesitation in recommending it be published in SciPost.

Recommendation

Publish (easily meets expectations and criteria for this Journal; among top 50%)

---

## Round 3 · Referee Report · Anonymous (Referee 2) · 2024-8-19

Report

The author has duly revised the paper considering the points raised, and in particular, improving the presentation. I now recommend the paper for publication.

Recommendation

Publish (easily meets expectations and criteria for this Journal; among top 50%)

---

## Round 3 · Referee Report · Anonymous (Referee 3) · 2024-8-27

Report

The revised version of the paper has satisfactorily addressed all the points raised by the reviewers. The new version is more clear and detailed and enables also the general physics reader to understand the motivations and to put in a better context the result of the analysis.

Recommendation

Publish (easily meets expectations and criteria for this Journal; among top 50%)

---

## Round 3 · Author Response

I would like to thank the 3 referees for their constructive comments and attention to detail which helped me substantially improve this manuscript. This new version comes with an improved structure and additional subsections that are mostly aimed at providing a better discussion of recent work on gravity in light of the primarily technical aspects discussed in the rest of the note. Section 3 is now reorganized and more explicitly devoted to the relation with the equations used by Oppenheim and collaborators. In 3.1, the general idea of the approach by Oppenheim is now discussed. The section 5, Discussion, is also substantially expanded following the recommendations of the first referee.

In addition to these major structural changes, I have tried to address as best I could the more specific concerns of the referees. The answers appear below.

Referee 1

major weakness

Does not review in sufficient detail the recent work of Oppenheim and co-workers, and its relation to previous work on semiclassical gravity.

-> I agree the initial note may have been discussing this approach too indirectly. I believe the new subsection 3.1 and substantially expanded discussion should fix this problem.

In addition to the major weakness discussed above, I have these requests for changes:

  1. Equation (1) is only one model of continuous measurement -- the other is a jump unravelling of a master equation. This is eventually mentioned, but only 2 pages later. It would be better not to hold the knowledgeable reader is suspense for so long.

-> I now say right away that jump version exist but that we do not consider them. I kept the remark which comes 2 pages later, and expanded it to emphasize the study of the jump case would be useful (but I think it is fair to want to leave it for later: the tools are different, and we cannot address everything in an article)

  1. Equation (1) uses the superoperator notation of the textbook [1] for the deterministic part but not for the stochastic part. Is there any particular reason to replace H with M?

-> The textbook by Wiseman and Milburn is the bible of everyone who ever worked on open quantum systems and measurement dynamics. While I still have the book on my bedside table (as I should), I confess I have started following the French heresy (of Pierre Rouchon and the likes) of replacing H with M. The reason is that H is already quite overloaded (Hamiltonian, Hilbert space) and rendering the subtle difference between latex fonts is difficult on a black board when discussing. The replacement, M, does not seem as overloaded and can stand for "measurement". These are, of course, quite subjective motivations.

  1. It is claimed (correctly) that Eq. (9) is the most general form of feedback, but it is not explained why this is so.

-> I now explain that this is the most general because the regularity of the signal only allows linear terms.

  1. The statement “but all objects are now implicitly z dependent” may be misleading. It is exactly the 3 objects (or sets of objects) mentioned in the preceding sentence that can have z-dependent.

-> Indeed, I corrected this misleading sentence.

  1. The paper needs a hyphens in compound adjectives, e.g. in z-dependent, measurement-induced etc. It also needs an n-dash in the adjectival phrase "hybrid quantum--classical".

-> I thank the referee for their sharp eye. I honestly did not know what the rules were, and thus I am glad to have learned them in passing! It should now be corrected everywhere.

  1. The phrase "the “diagonal” form" is used on page 4 a few paragraphs before it is explained what this means.

-> I added a fairly detailed remark (now the first remark post SME) to explain what diagonal means and related it to the history of the Lindblad equation.

  1. Below Eq.(32), what follows “noting that” is well-known in previous literature. If the author can find where this was first noted, at least in this context, and reference it, that would be of service to the community.

-> I genuinely have no idea who first noticed this. I rediscovered it (while being well aware of the fact that this must be well known). I am happy to quote a reference for it, but I do not really know what to look for (I am pretty sure such a fact must have been mentioned in passing, not as the main subject of an article)

  1. Regarding “Further, since the continuous measurement interpretation disappears, it is unclear if such models can be used as a basis for consistent hybrid dynamics”, could the author say something stronger here? Or at least expand upon the problem for those unfamiliar with it.

-> I expanded this discussion to make clearer the problem (essentially there is no equivalent of the signal, hence no quantum--classical glue.)

  1. With regard to the major issue mentioned in the report, here is an example of where more information would be highly desirable: “However, because the problematic quantum measurement part is left unaddressed ...” Why and how was this left unaddressed by Oppenheim and co-workers?

-> Precisely this part is expanded in the discussion and now has its own subsection before the conclusion.

Referee 2

Major weaknesses 1. The alternative approach in which jump stochastic processes (such as photodetection) replace weak continuous measurements (i.e. homodyne/heterodyne like measurements, with a strong local oscillator) is mentioned only briefly. It is not evident if this at the end does not change too much the physics or not. A more explicit comment on what can be achieved considering jump processes and eventually a more explicit comparison with Refs. 10 and 22 could be extremely useful to the reader.

  1. The derivation of the Markovian limit is a bit misleading in my opinion (see below) Requested changes

  2. The alternative approach based on jump stochastic processes should be discussed in more detail and not only almost incidentally as it is done in the present version. I understand that the mathematics can be different and one cannot exploit the effectiveness of the Ito calculus, but a more detailed discussion of the approach of Ref. 10 and 22 would provide a broader and clearer view.

-> I have chosen to be more explicit about the fact that the jump case is important, conceptually similar but possibly quantitatively distinct, and should be studied separately. I now discuss ref 10 in more detail, but mostly in its relation to the whole program of constructing theories of gravity. I left a more detailed discussion of 22 to future work (where one should discuss joint diffusive--jumpy quantum--classical dynamics). This is indeed a limitation, but the note is already almost 20 page long, and I think it is fair to ask for such a study to be postponed (although I do not question that it would be useful).

  1. The derivation of Sec IV of the Markovian limit in which the dynamics of the classical variables is adiabatically eliminated should be adjusted in my opinion. In fact, The equations, starting from eq. 24 seem to be written only for a single feedback term, i.e., a single feedback operator b_k. Instead I expect that, in order to be more general, we have many operators b_k, one for each measured signal r_k. However in most of the equations below the terms with b_k are always outside the sum over the index k. This makes the derivation misleading and I think it would be proper to adjust it, since the beginning, by assuming a general sum over k in the fluctuating potential of Eq. (24).

-> Yes I now understand why this part was unclear. I added explicit sums over k everywhere, and it should now be clear that the feedback is indeed the most general.

Referee 3

major weaknesses

  1. Connection to the literature, both in view of the possible physical applications and in view of the general formalism, is not adequate.

-> I hope the extended discussion in section 5 now makes the connection with literature clearer.

  1. The focus of the paper should be more clearly stated.

-> I now make it explicit that we deal only with continuous dynamics in the classical variables.

  1. In the presentation it is not always clear what is taken for granted and what is derived.

-> In the current version of the note, the starting point, the SME for the density matrix and the signal, is taken for granted as explained in the introduction. In principle, all the other equations are derived. If I missed something, and parts are inappropriately taken for granted, I am happy to clarify the derivation or more explicitly refer to the literature.

Requested changes

A better introduction to the addressed physical background.

-> The main contribution of this note is to clarify technical aspects, but I now expanded the discussion of the relation with physical theories, which now has a complete section (section 5 Discussion) with 3 subsections. I hope this addresses the physical background better, even if it comes after.

Clearer motivation for not dealing with jump noise and the possible new implications in considering it.

-> My motivation for not considering the jump case is mostly that it would add complexity in the presentation. In a secondary way, as argued in the remark, the possibility of continuous dynamics is less obvious (when starting from general hybrid dynamics), subjectively nicer for a fundamental theory, and thus why I was more attracted to discussing it. My fault was to not make that clear enough in the previous version. I am now more explicit that the jump case is not considered and that, although the same general reasoning remains valid in principle, the precise quantitative consequences deserve to be discussed. I grant that this is a limitation.

The abstract mentions considering non-Markovian feedback, but apparently, the point is not further discussed.

-> Apart from section 4, which discusses Markovian feedback, all the rest of the paper is fully non-Markovian (as far as feedback is concerned). In particular, all the content of sections 2 and 3 where the bulk of the mathematics is presented, the feedback is fully non-Markovian and general. Of course, it is more difficult to derive simple predictions from such general equations, which motivates section 4 which gives some results specific to the Markovian limit.

The wording "without additional positivity constraints" in the abstract is somehow deceitful to me. The stochastic differential equation and the corresponding Fokker-Planck are equivalent.

-> I agree with the referee that the SDE and Fokker-Planck PDE are indeed fully equivalent, and this is stated in the note. The point about positivity constraints is related the parameterization of the SDE/PDE. I think the new structure of section 3 makes this fact much clearer. I now show the diagonal and non-diagonal parameterization on the same PDE, to emphasize that this is not an issue of SDE vs PDE.

Summations over k appear and disappear and should be made more consistent in notation to avoid confusing the reader.

-> I agree, this was unclear. Now all summations on k are explicit. Einstein's summation is used only for the a,b index of classical variables.

---

## Round 3 · List of Changes

The article has been substantially revised, and the main change is thus a reshuffling and expansion of section 3 and 5. This is of course in addition with the specific answers to the requests of the referees mentioned above.

More specifically, section 3 is now reorganized and more explicitly devoted to the relation with the equations used by Oppenheim and collaborators. In particular, in 3.1, the general idea of the approach by Oppenheim is now explained. The section 5, Discussion, is also substantially expanded following the recommendations of the first referee.

---

## Editorial Decision

published